# Restoration Efficacy of *Picea likiangensis* var. *rubescens* Rehder & E. H. Wilson Plantations on the Soil Microbial Community Structure and Function in a Subalpine Area

**DOI:** 10.3390/microorganisms9061145

**Published:** 2021-05-26

**Authors:** Jixin Cao, Songlin Shi, Hong Pan, Zhan Chen, He Shang

**Affiliations:** 1Key Laboratory of Forest Ecology and Environment of National Forestry and Grassland Administration, Research Institute of Forest Ecology, Environment and Protection, Chinese Academy of Forestry, Beijing 100091, China; caojx@caf.ac.cn (J.C.); 15176369309@163.com (H.P.); chenzhan0508@126.com (Z.C.); 2College of Tourism and Urban-Rural Planning, Chengdu University of Technology, Chengdu 610059, China; shisonglin17@cdut.edu.cn

**Keywords:** community structure and function, soil microorganisms, subalpine area, reforestation, restoration ecology

## Abstract

The knowledge concerning the relationship between vegetation restoration and soil microorganisms is limited, especially at high altitudes. In order to evaluate the restoration efficacy of the reforestation on the soil microbial community, we have examined vegetation composition, edaphic properties and structure and function of different soil microbial groups in two different aged (25- and 40-year-old) *Picea likiangensis* var. *rubescens* Rehder & E. H. Wilson (*P*. *rubescens*) plantations and the primeval coniferous forest (PCF) dominated by *Abies squamata* Masters by plot-level inventories and sampling in western Sichuan Province, China. Our results suggested that only the fungal samples in 25-year-old *P*. *rubescens* plantation could be distinguished from those in the PCF in both structure and function. The structure and function of the fungal community recovered relatively slowly compared with bacterial and archaeal communities. In addition to the soil chemical properties and tree species composition, the shrub composition was also a key factor influencing the soil microbial community. The *P. rubescens* plantations were conducive to restoring the soil microbial community in both structure and function. However, there were uncertainties in the variations of the bacterial and archaeal communities with increasing the *P*. *rubescens* plantation age.

## 1. Introduction

The environmental degradation across the global terrestrial ecosystem leads to a decline in their biodiversity and function, while highlights the urgent need for ecological restoration [1,2,3,4]. Reforestation is an effective practice in ecological restoration and many countries have carried out large-scale plantation programs intending to mitigate land degradation, air pollution and climate change and these programs, especially in China, significantly reduce the net loss of global forests [5,6]. However, plantation programs primarily focus on the aboveground vegetation, in contrast, restorations of soil and belowground communities, particularly soil microbial communities, were often ignored [7]. Soil microbial communities are essential components of forest ecosystems because soil microorganisms ranging from free-living bacteria to single fungi play critical roles in the biogeochemical processes of the ecosystem [8,9,10,11]. Therefore, the restoration efforts with overlooking soil microbial community may fail to effectively restore the whole ecosystem.

Clear-cutting involving the felling and removal of all tree stems causes extreme changes in the biotic and abiotic environment and indirectly influences the distribution and structure of soil microbial community [12,13,14]. These extreme environmental conditions would restrict sensitive soil microbial species and reduce the microbial biomass and diversity [15,16]. Microbial communities are subject to the regulation of available resources, vegetation inputs, environmental stress and biological interaction [17]. Both soil and plant factors including soil pH, soil nutrient, soil water, soil texture and vegetation composition were reported to be the main drivers for the variation in soil microbial community [11,15]. Environmental restoration leading by reforestation may create new ecological niches and eventually contribute to the recovery of soil microbial structure and function [16,18,19]. However, the restoration efficacy of reforestation on the soil microbial structure and function is still uncertain due to the differences among ecosystems or the complex interaction between biotic and abiotic factors [20,21,22]. Additionally, different soil microorganisms may have distinct responses to the same environmental changes [23]. Therefore, it is necessary to carry out studies on the different microbial groups in typical areas to better understand the impact of reforestation on soil microorganisms.

The subalpine forest area of western Sichuan is located in the transition zone between the Tibetan Plateau and Sichuan Basin in China. The primeval coniferous forests dominated by fir (*Abies* spp.) and spruce (*Picea* spp.) in this area play a critical role in reducing soil erosion, regulating runoff, conserving water sources and biodiversity in the upper reaches of the Yangtze River [24,25,26]. Unfortunately, these primeval coniferous forests have suffered long-term exploitation during the period of the mid-20th to the end of the 20th century. After clear-cutting, spruce (*Picea* spp.) was commonly used to implement reforestation on the cutovers because of its adaptability to harsh climate conditions and its fast growth [24,27]. Changes in aboveground vegetation would significantly affect soil microbial communities by the differences in root and exudates, leaf litter, or edaphic properties [28]. However, little attention was paid to the relationship between the spruce plantation and soil microbial community and whether the spruce plantations have a restoration effect on soil microbial structure and function in this subalpine area. We hypothesis that (1) the *Picea likiangensis* var. *rubescens* Rehder & E. H. Wilson (*P. rubescens*) plantations established on cutovers would be beneficial to restoring the soil microbial community and (2) different soil microbial groups would follow distinct restoration trajectories. Our study has verified the above hypotheses and attempted to comprehensively analyze the differences in plants, soil and microbial communities between *P. rubescens* plantations and the primeval coniferous forest (PCF). This study could provide a clue to underlying the influence mechanisms of spruce plantations on soil microorganisms in subalpine areas.

## 2. Materials and Methods

### 2.1. Study Sites

The present study was conducted in a state-owned forest farm (30°41′−30°42′ N, 101°11′−101°14′), which is located in Daofu County, western Sichuan Province, China. This area belongs to the continental plateau monsoon climate with an annual mean temperature of around 11.0 °C and annual precipitation of 550–600 mm [29]. Two different aged *P. rubescens* plantations, namely, 25 and 40 years old, were chosen to compare with the PCF dominated by *Abies squamata* Masters (*A.squamata*). These *P. rubescens* plantations were all established on cutovers, which were primeval coniferous forests before clear-cutting. All stands are on the shady slope of subalpine (Table 1). The soils of the three different stands are all classified as a mountain brown soil series in Chinese soil taxonomy and the parent materials are mainly phyllite and slate. Moreover, there were little litters and organic matters in the surface layer of soils in these three different stands because of the steep slope. Due to the excessive initial planting density, the spruce plantations could be self-thinning during the stand development and then the number of broad-leaved tree species (e.g., *Betula albosinensis* Burkill and *Betula utilis* D. Don) would gradually increase [30]. In the present study, the average number ratio of broad-leaved trees in 25-year-old *P. rubescens* plantation, 40-year-old *P. rubescens* plantation and PCF was 1%, 21% and 45%, respectively. The mean coverage of trees in the 25-year-old *P. rubescens* plantation, 40-year-old *P. rubescens* plantation and the PCF was 67%, 75% and 45%, respectively. Shrubs in the *P. rubescens* plantations are dominated by *Sorbus vilmorinii* Schneid and *Rosa omeiensis* Rolfe, while shrubs in the PCF are dominated by *Rhododendron vernicosum* Franch and *Cerasus clarofolia* (Schneid.) Yü et Li. The mean coverage of shrubs in the 25-year-old *P. rubescens* plantation, 40-year-old *P. rubescens* plantation and the PCF was 37%, 66% and 70%, respectively. Herbs in the plantations are dominated by *Parasenecio latipes* (Franch.) Y.L. Chen, *Viola biflora* var. *rockiana* (W. Becker) Y.S. Chen and *Athyrium mackinnonii* (Hope) C. Chr., while herbs in the PCF are dominated by *Parasenecio latipes* and *Allium ovalifolium* Hand.-Mzt. The mean coverage of herbs in the 25-year-old *P.*
*rubescens* plantation, 40-year-old *P. rubescens* plantation and the PCF was 24%, 26% and 48%, respectively.

### 2.2. Sampling Design

The field samplings and measurements were carried out in July 2019. Three 20 × 20 m plots were randomly established in each stand. All tree species and corresponding numbers within each plot were identified and recorded, respectively. Three 5 × 5 m subplots were randomly distributed within each plot for shrub species diversity determination. Moreover, there was a micro plot for herb species diversity determination in each subplot. Five soil samples that were near or in the micro plots were randomly collected at a depth of 0–10 cm within each plot and these soil samples were then combined to create one composite soil sample. After fully mixing, approximately 500 g composite soil sample was obtained for each plot. All the composite soil samples were held on ice in an incubator. After removing the rocks, roots and litters, the soil samples were passed through a 2 mm sieve and then each soil sample was divided into two parts where one was used for soil chemical properties analysis and the remaining part was used for the extraction of DNA (stored at −80 °C). Furthermore, three soil samples of 0–10 cm layer were also randomly collected within each plot using a 100 cm^3^ soil ring for soil maximum water holding capacity (MWHC) and bulk density (BD) determination. All these soil samples were also near or in the micro plots.

### 2.3. Edaphic Properties and Plant Diversity Analysis 

Soil organic carbon (SOC) concentration was determined by the potassium dichromate oxidation method and total nitrogen (TN) concentration was determined by the semi micro-Kjeldahl method [31]. The C/N ratio was calculated by dividing the SOC concentration by the corresponding TN concentration. Soil ammonium nitrogen (AN) and nitrate nitrogen (NN) concentrations were determined with an AA3 Continuous Flow Analytical System (Bran+Luebbe, German). Soil available potassium (AK) concentration was determined by the method of flame photometer [31]. Soil available phosphorus (AP) concentration was determined by the Mo-Sb colorimetric method [31]. Soil pH was determined by an FE20-K pH meter (Mettler Toledo, Switzerland) in carbon-dioxide-free water: soil ratio of 5: 1(V/W) suspensions. Soil MWHC was determined by the method of immersion drying [32]. Soil samples with soil rings were oven-dried at 105 °C to constant weight for determining soil BD. The species diversity of the plant was represented by the Shannon index calculated based on the abundance ratio of each species. The mean of three corresponding replications represented the soil MWHC, soil BD, shrub and herb species diversity in each plot, respectively.

### 2.4. DNA Extraction, Library Construction and Metagenomic Sequencing

DNA for metagenomic analysis was extracted from approximately 0.5 g soil samples using the fastDNA Spin Kit (MP Biomedicals, Solon, OH, USA). The concentration and purity of extracted DNA were determined using a TBS-380 mini-Fluorometer (Turner Biosystems, Sunnyvale, CA, USA) and an ND-2000 UV-Vis spectrophotometer (Thermo Fisher Scientific-NanoDrop, Wilmington, DE, USA), respectively. Extracted DNA was assessed using 1% agarose gel electrophoresis. DNA was sheared into fragments of approximately 400 bp using an M220 Focused-ultrasonicator (Covaris, Woburn, MA, USA). DNA templates were then treated using the Rapid DNA-Seq Kit (Bioo Scientific, Austin, TX, USA) to build a paired-end library. Pair-end sequencing (2 × 150 bp) was completed on the Novaseq platform (Illumina, San Diego, CA, USA) by Shanghai Majorbio Biopharm Technology Co., Ltd. in Shanghai, China. 

### 2.5. Metagenome Sequence Assembly, Gene Prediction, Taxonomy and Functional Assignment

The 3′-adaptors and 5′-adaptors were stripped using fastp (https://github.com/OpenGene/fastp, version 0.20.0 accessed on 15–30 November 2019) [33,34]. The raw reads with a minimum quality score of 20 and a minimum length of 50 bp were quality processed [34]. Clean reads were assembled using MEGAHIT (https://github.com/voutcn/megahit, version 1.1.2, accessed on 15–30 November 2019) [35]. Contigs longer than 300 bp were selected as the final assembly result [35]. The open reading frames (ORFs) of the selected contigs were predicted with MetaGene (http://metagene.cb.k.u-tokyo.ac.jp/, accessed on 15–30 November 2019) [36]. The predicted ORFs longer than or equal to 100 bp were retained and translated into amino acid sequences using NCBI ORF finder (https://www.ncbi.nlm.nih.gov/orffinder/, accessed on 15–30 November 2019). A nonredundant gene catalog was constructed using CD-HIT (http://www.bioinformatics.org/cd-hit/, version 4.6.1, accessed on 15–30 November 2019) with the sequence identity cutoff of 0.9 and a minimum coverage cutoff of 0.9 [35]. The high-quality reads of the sequence data from each sample were compared with the nonredundant gene catalog to obtain the gene set using SOAP align (http://soap.genomics.org.cn/, version 2.21, accessed on 15–30 November 2019) with the identity of 0.95 [37]. The microbial profile of structure and functions were generated by Diamond (http://www.diamondsearch.org/index.php, version 0.8.35, accessed on 15–30 November 2019) through comparison with the NCBI NR and Kyoto Encyclopedia of Genes and Genomes (KEGG) database with an e-value cutoff of 1 × 10^−5^, respectively [34,35,38]. All these application programs were accessed on 15–30 November 2019. Moreover, we created bacterial, fungal and archaeal gene catalogs based on the initial nonredundant gene catalog for further analysis.

### 2.6. Statistical Analysis 

In the present study, α-diversity was used to analyze the complexity of the soil microbial diversity of species and functions based on Shannon index in Mothur (http://www.mothur.org/wiki accessed on 15–30 November 2019) [39], which was accessed on 30 November 2021. Principal coordinate analysis (PCoA) and analysis of similarities (ANOSIM) based on the Bray–Curtis distance were applied to determine differences in the microbial community among the three stands using the R Vegan package [40]. Mantel tests analysis was used to determine the contribution of the related environmental variables to soil microbial species assembly and functions using the R Vegan package [41]. Moreover, a phylum with a relative abundance of greater than 1% in all samples was defined as a dominant phylum in the present study. 

To assess the relative importance of selection and chance effects, the deterministic and stochastic changes were calculated as variations in community structure using a modified method based on Euclidean distances [42,43]. Compared with the 40-year-old *P. rubescens* plantation, the 25-year-old *P. rubescens* plantation is in an earlier development stage. Therefore, the variation between the plots of the 25-year-old *P. rubescens* plantation was used as a reference point and the Euclidean distances were calculated based on the relative abundance of microbial communities at the class level and the relative abundance of functional gene categories at KEGG level 3. The deterministic change can be calculated as D = [(mean variation between the reference point and the other stand)(reference point)] and the stochastic change can be calculated by S = [(mean variation within the other stand) − (reference point)]. For the MF and PCF stands, we calculated the importance of chance = |S|/(|D| + |S|). If the importance of chance is less than 0.5, the deterministic change is dominant and if it is greater than 0.5, the stochastic change is dominant. For the reference point, the importance of chance was one. 

After checking for normality and homogeneity, the difference among the same variables in the three stands was examined by one-way analysis of variance (ANOVA) with Tukey’s test (*p* < 0.05) (SPSS, Chicago, IL, USA, version 18.0).

## 3. Results

### 3.1. Variations in the Microbial Community Structure

The bacterial communities were mainly composed of Proteobacteria, Acidobacteria, Actinobacteria, Bacteroidetes and Verrucomicrobia at the phylum level across all the stands (Figure 1a). The mean relative abundances of these 5 dominant phyla were 50.14%, 25.49%, 9.39%, 3.75% and 3.03%, respectively (Figure 1a). For the fungal communities, Basidiomycota and Ascomycota were the two dominant phyla and both of them accounted for more than 90% in all the stands (Figure 1b). There were also 5 dominant phyla within the archaeal communities and the mean relative abundance of the individual dominant phyla across all the stands was as follows: Euryarchaeota (69.92%), Thaumarchaeota (12.19%), Unclassified_d_Archaea (6.23%), Candidatus_Bathyarchaeota (4.66%) and Crenarchaeota (4.36%) (Figure 1c). Compared with the PCF, the numbers of the order with significant variation in the relative abundance decreased from 4 and 8 in the 25-year-old *P. rubescens* plantation to 1 and 5 in the 40-year-old *P. rubescens* plantation for the bacterial and fungal communities, respectively (*p* < 0.05) (Appendix A). In contrast, the number of the order with significant variation in the relative abundances increased from 4 in the 25-year-old *P. rubescens* plantation to 5 in the 40-year-old *P. rubescens* plantation for the archaeal community (*p* < 0.05) (Appendix A). 

The Shannon indices of these three different soil microbial groups were not significantly different among the three stands (Table 2). The patterns of bacterial, fungal and archaeal community β-diversity at the species level are shown in the PCoA plots (Figure 2). Only the fungal communities were differentiated into three clusters corresponding to the three different stands (Figure 2b). Furthermore, the *r*-value of the bacterial and fungal community structure across the three different stands based on ANOSIM analysis reached 0.35 and 0.88 (*p* < 0.05), respectively (Table 3).

### 3.2. Variations in the Microbial Community Function

The functional profiles of KEGG level 3 were compared among the three different stands and the relative abundance of the functional gene categories with a significant difference between the plantations and the PCF were listed in Table 4. Compared with the PCF, the number of functional gene categories with a significant variation in the relative abundance for the bacterial, fungal and archaeal communities decreased from 5, 11 and 7 in the 25-year-old *P. rubescens* plantation to 0, 6 and 2 in the 40-year-old *P. rubescens* plantation, respectively (*p* < 0.05) (Table 4). For the functional gene categories of bacteria and fungi in Table 4, the relative abundances of all the functional gene categories related to metabolism in the 25-year-old *P. rubescens* plantation were significantly higher than the corresponding relative abundances in the PCF (*p* < 0.05). These functional gene categories included ABC transporters, glyoxylate and dicarboxylate metabolism, methane metabolism, propanoate metabolism, oxidative phosphorylation 2-oxocarboxylic acid metabolism, ether lipid metabolism and lysine biosynthesis (Table 4). However, the relative abundances of the functional gene categories related to the archaeal metabolism did not show a consistent variation trend in the 25-year-old *P. rubescens* plantation compared with the PCF (Table 4). Regarding the fungal functional gene categories related to the cellular processes and genetic information processing, the relative abundances of cell cycle–yeast, spliceosome, RNA transport, cell cycle, meiosis–yeast and autophagy–yeast in 25-year-old *P. rubescens* plantation were significantly lower than the corresponding relative abundances in the PCF (*p* < 0.05) (Table 4). 

According to the Shannon index, the functional α-diversity of the fungal community in the 25-year-old and 40-year-old *P. rubescens* plantation significantly decreased by 35% and 27% compared with that in the PCF, respectively (Table 5). In contrast, the Shannon index of both the bacterial and archaeal community varied little among the three different stands (Table 5). The cluster of functional modules of the fungal community in the 25-year-old *P. rubescens* plantation was differentiated from that in both the 40-year-old *P. rubescens* plantation and the PCF (Figure 3b). However, the clusters were difficult to be distinguished according to the different stands for both bacterial and archaeal community functions (Figure 3a,c). In addition, only the p-value corresponding to the fungal community functions across three different stands was lower than 0.05 based on ANOSIM (Table 6).

### 3.3. The Relationship between the Soil Microbes and Environmental Variables 

Table 7 shows the variations in the environmental variables within the three different stands. The values of soil AP concentration, AK concentration and pH in the two *P. rubescens* plantations were significantly higher than the corresponding value in the PCF (*p* < 0.05) (Table 7). In contrast, the soil NN concentration in the 40-year-old *P. rubescens* plantation was not significantly different from that in the PCF (*p* < 0.05) (Table 7). Regarding the α-diversity of the plant, the α-diversity of the trees and shrubs in the 25-year-old *P. rubescens* plantation was significantly lower than that in the other two stands (*p* < 0.05) (Table 7). Additionally, the α-diversity of the herbs in the 40-year-old *P. rubescens* plantation was slightly higher than that in the PCF (*p* < 0.05) (Table 7).

Table 8 shows that the variation in the soil bacterial community structure was only significantly related to the tree α-diversity among the environmental variables (*p* < 0.05). In addition to the tree α-diversity, the soil NN, AK and shrub α-diversity were also significantly related to the function variation of the soil bacterial community (*p* < 0.05) (Table 9). In terms of soil fungal community structure and function, both of them were significantly correlated with the soil NN, C/N, pH, tree α-diversity and shrub α-diversity (*p* < 0.05) (Table 8 and Table 9). However, no environmental variables were significantly associated with the variation in soil archaeal community structure and function (Table 8 and Table 9). Using the 25-year-old *P. rubescens* plantation as a reference point, the changes in the fungal community structure and function were dominated by a deterministic process in the 40-year-old *P. rubescens* plantation (Figure 4). The variations in archaeal community structure and the bacterial community function were also dominated by the deterministic processes in the 40-year-old *P. rubescens* plantation (Figure 4). However, the variations in the bacterial community structure and the function of the archaeal community were dominated by the stochastic process (Figure 4).

## 4. Discussion

Although the dominant phyla of the three different microbial groups were similar among the three stands in the present study, the relative abundance of many microbial orders significantly changed in the two *P. rubescens* plantations compared with the PCF. These variations in community structure might be a dominant response of soil microorganisms to the changes in environmental conditions [44]. Our results reveal that the primeval forests are more suitable for some specific fungi, such as *Russulale*, *Polyporales, Gloeophyllales* and *Corticiales*. Especially, the relative abundance of *Russulale* in the two *P. rubescens* plantations was far less than that in the PCF, which is similar to a previous finding on the natural vegetation restoration in a subalpine area [40]. This phenomenon may be strongly associated with the differences in the saprophytic conditions between the primeval forests and the plantations. At the order level, the relative abundance of *Solirubrobacterales*, *Pseudonocardiales* and *Micrococcales* were significantly higher in the 25-year-old *P. rubescens* plantation than in the PCF. Additionally, the relative abundance of *Rhizobiales* in the two *P. rubescens* plantations was also significantly higher than that in the PCF. These results are consistent with the other studies that demonstrated that the members of *Actinobacteria* and *Proteobacteria* are commonly be regarded as copiotrophic microbes enriched in the soil of high nutrient availability [45,46,47,48]. In the subalpine areas of western Sichuan, previous studies showed that clear-cutting would improve the soil hydrothermal conditions, which enhanced the mineralization of organic matter and then increased soil pH and soil nutrient availability [49,50]. We suggest all these changes lead to an increase in the relative abundance of copiotrophic microorganisms.

In the present study, we found the variation in the soil bacterial community structure was only significantly correlated with the tree species α-diversity. Plants and plant communities could affect the bacterial community structure by changing soil properties and habitats mainly through root growth, the production of litter and root exudates [51,52,53]. However, there was little litter on the surface of mineral soil in our study sites because of the steep slope. Hence, we suggest the metabolism of the tree root is the main driver for the change of the bacterial community structure in this area. In contrast, the variation in the soil fungal community structure was simultaneously related to soil C/N, NN, pH, tree α-diversity and shrub α-diversity, which indicates the fungal community was more sensitive to the variations in the environmental variables than the bacterial community. This result is consistent with other studies, which can be explained by that fungi may outcompete other soil microbial groups during the stand development because they utilize the C available from plants more effectively and they could establish a close association with plants, i.e., the development of mycorrhizal [37,54,55]. Additionally, bacteria are generally more resilient in the face of disturbances because of their relatively high growth rate [56]. These inconsistent results between the bacterial and fungal communities indicate that the key influencing factors could be different for the various soil microbial groups during the restoration process. Notably, the fungal community structure was significantly correlated with the shrub species composition. This result may be associated with the host specificity of some mycorrhizal [57]. Therefore, the increased plant richness could increase the soil fungal diversity and the *P. rubescens* plantation with an appropriate arrangement of shrub species would be more conducive to the recovery of the fungal community in this area.

Understanding microbial functional group response to changing environmental variables is essential for the comprehensive evaluation of the ecological restoration efficacy [17,45]. In the present study, we found that the bacterial and fungal gene sequences related to the metabolism were more abundant in the 25-year-old *P. rubescens* plantation than those in the PCF. Additionally, the relative abundance of alanine, aspartate and glutamate metabolism of the archaeal community in the 25-year-old *P. rubescens* plantation was also significantly higher than that in the PCF. These results are suggested to be associated with the higher nutrient availability after clear-cutting because the higher nutrient availability is more favorable for soil microbial activities and energy metabolism [58,59]. Enhancement of metabolic activities would contribute to accelerating organic matter degradation and nutrient cycling [56]. Our results also showed the relative abundance of cell cycle-yeast, spliceosome, RNA transport, cell cycle-yeast, meiosis-yeast and autophagy-yeast in the fungal community in the 25-year-old *P. rubescens* plantation was significantly lower than those in the PCF. These changes related to cellular processes and genetic information processing indicate the rate of fungal growth and turnover in the early *P. rubescens* plantation may be slower than that in the PCF. Decreasing the rate of microbial growth and turnover would reduce the rate of soil organic carbon accumulation to some degree [60]. Moreover, the relative abundance of some specific functional gene categories of the fungal community in the 40-year-old *P. rubescens* plantation increased to a level similar to that in the PCF, indicating the functional pattern of the fungal community would gradually converge to that of the PCF with increasing plantation age.

The present study showed that the soil chemical properties, tree diversity and shrub diversity played prominent roles in regulating the functional pattern of bacterial and fungal communities. Different from the bacterial community structure, the functional pattern of the bacterial community was strongly associated with soil available potassium. Previous studies suggested that soil bacteria can affect the solubility and availability of potassium through a range of processes, including chelation, redox, acidolysis and production of different products, which conversely affects the filtering of bacterial species related to potassium and then influence the bacterial community structure and function [61,62]. Other key influencing factors such as pH and nitrate nitrogen could also regulate the functional patterns through the selection of microbial species [44,63], especially for the fungal community in the present study. Because we found that the key drivers influencing the fungal community structure and function were identical and both the variations in fungal community structure and function showed a deterministic process with the plantation age increasing. In contrast, the structural and functional changes in bacterial communities were driven by a different process. This result was similar to the previous studies that reported the structural and functional changes in the microbial community may diverge during the stand development [37,64].

## 5. Conclusions

There were obvious differences in the soil microbial community between the earlier stage of *P. rubescens* plantation and the PCF, which did not only show in both bacterial and fungal community structure but also the function of the fungal community. However, these differences were diminishing with increasing the *P. rubescens* plantation age. The structure and function of the fungal community recovered relatively slowly compared with those of bacterial and archaeal communities. Only the fungal community was dominated by the deterministic process in both structural and functional variation during the *P. rubescens* plantation development. In addition to the soil chemical properties and tree species composition, the shrub species composition was also a key factor for shaping the structure and function of the soil microbial community. The *P. rubescens* plantation with an appropriate shrub arrangement would be better for the recovery of the soil microbial community in the subalpine area. The information from this study can improve the understanding of the effects of reforestation on the belowground ecosystem in the subalpine area. Nevertheless, because of the limitation in the selection of stands, the results do not include the whole growth process of the *P. rubescens* plantation. Thus, future studies for the effects of stand ages on the soil microbial structure and function will be required to further expand our understanding.

## Figures and Tables

**Figure 1 microorganisms-09-01145-f001:**
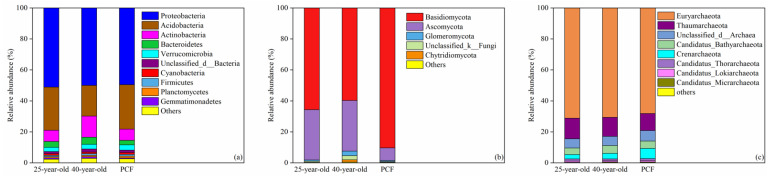
Taxonomic proportions in the bacterial (**a**), fungal (**b**) and archaeal (**c**) communities at the phylum level. 25-year-old, 40-year-old and PCF represent the 25-year-old *P. rubescens* plantation, 40-year-old *P. rubescens* plantation and the primeval coniferous forest, respectively. Others represent the sum of phyla that ccount for less than 1% in relative abundance.

**Figure 2 microorganisms-09-01145-f002:**
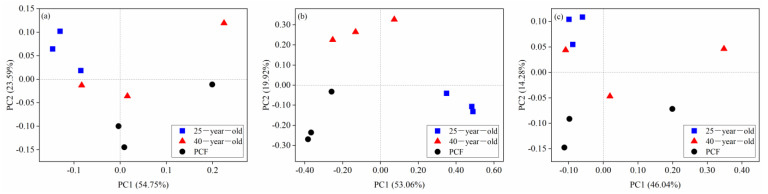
Principal coordinate analysis (PCoA) of the taxonomic patterns of bacterial (**a**), fungal (**b**) and archaeal (**c**) communities across the three stands at the species level. The 25-year-old, 40-year-old and PCF represent the 25-year-old *P. rubescens* plantation, 40-year-old *P. rubescens* plantation, and primeval coniferous forest, respectively.

**Figure 3 microorganisms-09-01145-f003:**
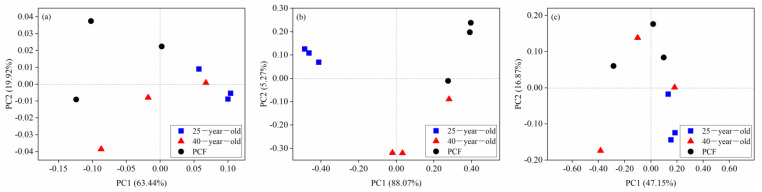
Principal coordinate analysis (PCoA) of the functional patterns of bacterial (**a**), fungal (**b**) and archaeal (**c**) communities across the three stands at the module level of KEGG. 25-year-old, 40-year-old and PCF represent the 25-year-old *P. rubescens* plantation, 40-year-old *P. rubescens* plantation and primeval coniferous forest, respectively.

**Figure 4 microorganisms-09-01145-f004:**
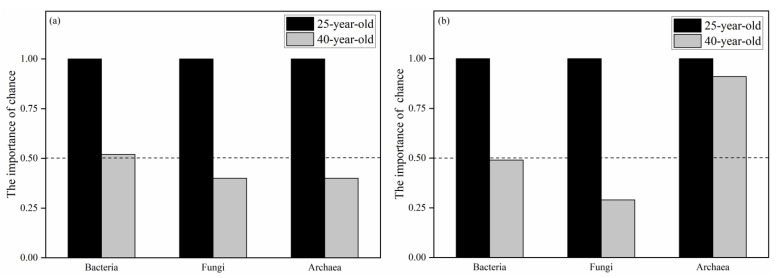
The relative importance of chance effect in the structure (**a**) and funtion (**b**) of bacterial, fungal and archaeal communities.

**Table 1 microorganisms-09-01145-t001:** Basic characteristics of the three different stands.

Stand	Dominated Tree Species	Elevation (m a. s. l.)	Slope Degree (°)	Slope Aspect	Stand Density (Tree hm^−2^)	Mean DBH (cm)
25-year-old plantation	*Picea rubescens*	3718–3740	12–26	Shady	1833	11.83
40-year-old plantation	*Picea rubescens*	3728–3760	14–32	Shady	1433	20.61
Primeval coniferous forest	*Abies squamata*	3770–3802	14–26	Shady	741	24.00

DBH, diameter at breast height.

**Table 2 microorganisms-09-01145-t002:** Taxonomic α-diversity of the microbial community in the three stands at the species level.

	25-Year-Old	40-Year-Old	PCF
Bacteria	6.59 (±0.05) a	6.23 (±0.15) a	6.29 (±0.07) a
Fungi	4.62 (±0.15) a	3.90 (±0.27) a	3.86 (±0.17) a
Archaea	5.38 (±0.02) a	5.18 (±0.07) a	5.28 (±0.03) a

Data are presented as the mean value with the standard error (SE) given in parenthesis (*n* = 3). Mean values within a row followed by different lowercase letters are significantly different at *p* < 0.05. Data analysis was based on one-way analysis of variance (ANOVA) with Tukey’s test (*p* < 0.05). The 25-year-old, 40-year-old and PCF represent the 25-year-old *P. rubescens* plantation, 40-year-old *P. rubescent* plantation and the primeval coniferous forest, respectively. The α-diversity is represented by Shannon index.

**Table 3 microorganisms-09-01145-t003:** Structural differences of microbial community among the three stands at the species level.

Bacteria		Fungi		Archaea	
*r*	*p*	*r*	*p*	*r*	*p*
0.35 *	0.045	0.88 **	0.007	0.20	0.084

* *p* < 0.05, ** *p* < 0.01. Data analysis was based on ANOSIM.

**Table 4 microorganisms-09-01145-t004:** The differences in relative abundance (%) of the soil microbial functional gene categories in the three stands.

	KEGG Pathway Level 3	25-Year-Old	40-Year-Old	PCF
Bacterial community	ABC transporters	3.31 (±0.11) a	3.14 (±0.08) ab	2.82 (±0.07) b
	Two-component system	2.66 (±0.03) a	2.93 (±0.09) ab	3.02 (±0.08) b
	Glyoxylate and dicarboxylate metabolism	1.70 (0.02) a	1.66 (±0.02) ab	1.59 (±0.04) b
	Methane metabolism	1.42 (±0.03) a	1.30 (±0.03) b	1.25 (±0.02) b
	Propanoate metabolism	1.36 (±0.01) a	1.21 (±0.04) b	1.19 (±0.03) b
Fungal community	Oxidative phosphorylation	9.80 (±1.28) a	9.83 (±2.40) a	2.48 (±0.44) b
	Cell cycle-yeast	0.32 (±0.08) a	0.98 (±0.20) b	1.83 (±0.14) c
	Spliceosome	0.22 (±0.09) a	1.19 (±0.27) ab	1.38 (±0.06) b
	RNA transport	0.20 (±0.07) a	0.84 (±0.24) ab	1.50 (±0.03) b
	MAPK signaling pathway-yeast	0.34 (±0.03) a	0.55 (±0.11) a	1.50 (±0.10) b
	Ribosome biogenesis in eukaryotes	0.78 (±0.15) ab	0.68 (±0.18) a	1.37 (±0.11) b
	Cell cycle	0.32 (±0.05) a	0.79 (±0.16) a	1.36 (±0.11) b
	Meiosis-yeast	0.10 (±0.07) a	0.73 (±0.26) ab	1.30 (±0.13) b
	Autophagy-yeast	0.23 (±0.10) a	0.65 (±0.15) ab	0.92 (±0.05) b
	2-Oxocarboxylic acid metabolism	1.41 (±0.13) a	0.46 (±0.14) b	0.60 (±0.03) b
	Ether lipid metabolism	2.18 (±0.73) a	0.47 (±0.20) ab	0.18 (±0.04) b
	Lysine biosynthesis	0.94 (±0.11) a	0.13 (±0.01) b	0.27 (±0.01) c
Archaeal community	Amino sugar and nucleotide sugar metabolism	1.66 (±0.14) a	1.91 (±0.25) a	3.36 (±0.25) b
	Alanine, aspartate and glutamate metabolism	2.62 (±0.28) a	1.35 (±0.05) b	1.51 (±0.10) b
	Glycolysis/Gluconeogenesis	2.19 (±0.09) a	1.86 (±0.31) ab	1.31 (±0.12) b
	Two-component system	1.48 (±0.08) a	1.84 (±0.25) ab	2.22 (±0.06) b
	Pantothenate and CoA biosynthesis	1.60 (±0.14) ab	2.01 (±0.22) a	1.07 (±0.11) b
	Starch and sucrose metabolism	0.95 (±0.08) a	1.11 (±0.11) ab	1.34 (±0.04) b
	Fructose and mannose metabolism	0.80 (±0.05) a	1.28 (±0.11) b	1.45 (±0.06) b
	Biofilm formation-Vibrio cholerae	0.20 (±0.04) a	0.38 (±0.08) ab	0.79 (±0.22) b

Data are presented as the mean value with the standard error (SE) given in parenthesis (*n* = 3). Mean values within a row followed by different lowercase letters are significantly different at *p* < 0.05. Data analysis was based on one-way analysis of variance (ANOVA) with Tukey’s test (*p* < 0.05). The 25-year-old, 40-year-old and PCF represent 25-year-old *P. rubescens* plantation, 40-year-old *P. rubescens* plantation and primeval coniferous forest, respectively.

**Table 5 microorganisms-09-01145-t005:** Functional α-diversity of the microbial community in the three stands at the module level of KEGG.

	25-Year-Old	40-Year-Old	PCF
Bacteria	4.95 (±0.00) a	4.99 (±0.02) a	4.99 (±0.02) a
Fungi	2.99 (±0.17) a	3.36 (±0.37) a	4.61 (±0.06) b
Archaea	4.09 (±0.01) a	4.03 (±0.04) ab	4.01 (±0.01) b

Data are presented as the mean value with the standard error (SE) given in parenthesis (*n* = 3). Mean values within a row followed by different lowercase letters are significantly different at *p* < 0.05. Data analysis was based on one-way analysis of variance (ANOVA) with Tukey’s test (*p* < 0.05). The 25-year-old, 40-year-old and PCF represent the 25-year-old *P. rubescens* plantation, 40-year-old *P. rubescens* plantation and the primeval coniferous forest, respectively. The α-diversity is represented by the Shannon index.

**Table 6 microorganisms-09-01145-t006:** Functional differences of microbial community among three stands at the module level of KEGG.

Bacteria		Fungi		Archaea	
*r*	*p*	*r*	*p*	*r*	*p*
0.38	0.090	0.89 **	0.007	0.20	0.132

** *p* < 0.01. Data analysis was based on ANOSIM.

**Table 7 microorganisms-09-01145-t007:** Variation in the edaphic and biotic factors in the three stands.

		25-Year-Old	40-Year-Old	PCF
Edaphic factors	SOC (g kg^−1^)	102.98 (±5.22) A	124.41 (±10.74) A	103.27 (±17.91) A
	TN (g kg^−1^)	6.14 (±0.27) A	6.19 (±0.47) A	5.86 (±0.91) A
	C/N	16.77 (±0.14) A	20.07 (±0.56) B	17.55 (±0.27) A
	AN (mg kg^−1^)	28.34 (±7.18) A	15.02 (±3.87) A	32.83 (±12.05) A
	NN (mg kg^−1^)	7.82 (±1.31) A	1.41 (±3.18) B	0.78 (±0.54) B
	AP (mg kg^−1^)	15.58 (±2.15) AB	18.37 (±2.92) A	7.50 (±1.32) B
	AK (mg kg^−1^)	397.67 (±42.48) A	390.33 (±69.93) A	160.33 (±26.64) B
	pH	5.65 (±0.03) A	5.47 (±0.17) A	4.79 (±0.06) B
	BD (g cm^−3^)	0.67 (±0.02) A	0.48 (±0.05) B	0.60 (±0.04) AB
	MWHC (*v v*^−1^, %)	34.58 (±2.46) A	27.42 (±1.31) A	35.75 (±2.63) A
Biotic factors	PR	22 (±2) A	27 (±2) A	23 (±3) A
	Tree α-diversity	0.05 (±0.02) A	0.73 (±0.09) B	0.78 (±0.12) B
	Shrub α-diversity	0.74 (±0.13) A	1.31 (±0.08) B	1.60 (±0.04) C
	Herb α-diversity	2.04 (±0.07) AB	2.26 (±0.04) B	1.92 (±0.11) A

Data are presented as the mean value with the standard error (SE) given in parenthesis (*n* = 3). Mean values within a row followed by different uppercase letters are significantly different at *p* < 0.05. 25-year-old, 40-year-old and PCF represent 25-year-old *P. rubescens* plantation, 40-year-old *P. rubescens* plantation and primeval coniferous forest, respectively. Data analysis was based on one-way analysis of variance (ANOVA) with Tukey’s test (*p* < 0.05). SOC soil organic carbon, TN total nitrogen, C/N the ratio of organic carbon to total nitrogen, AN ammonium-nitrogen, NN nitrate-nitrogen, AP available phosphorus, BD bulk density, MWHC maximum water holding capacity, PR plant richness, α-diversity was represented by Shannon index.

**Table 8 microorganisms-09-01145-t008:** The relationships between the environmental variables and the soil microbial community structure based on mantel test.

	Soil Bacteria		Soil Fungi		Soil Archaea	
Variables	*r*	*p*	*r*	*p*	*r*	*p*
SOC (g kg^−1^)	−0.204	0.924	−0.287	0.990	−0.211	0.954
TN (g kg^−1^)	−0.146	0.800	−0.342	1.000	−0.118	0.730
C/N	0.052	0.381	0.404	0.026	0.106	0.295
AN (mg kg^−1^)	−0.281	0.915	−0.269	0.986	−0.245	0.863
NN (mg kg^−1^)	0.198	0.152	0.586	0.010	−0.072	0.559
AP (mg kg^−1^)	−0.040	0.587	−0.044	0.588	0.036	0.358
AK (mg kg^−1^)	0.304	0.063	0.204	0.102	0.141	0.275
pH	0.051	0.427	0.317	0.042	−0.017	0.627
BD (g cm^−3^)	0.251	0.109	0.207	0.098	0.371	0.070
MWHC (*v v*^−1^, %)	−0.220	0.882	0.015	0.401	−0.056	0.507
PR	0.020	0.385	−0.137	0.780	−0.115	0.764
Tree α-diversity	0.407	0.035	0.651	0.003	0.109	0.243
Shrub α-diversity	0.239	0.149	0.701	0.001	−0.047	0.417
Herb α-diversity	−0.122	0.706	−0.110	0.731	−0.126	0.720

SOC soil organic carbon, TN total nitrogen, C/N the ratio of organic carbon to total nitrogen, AN ammonium nitrogen, NN nitrate nitrogen, AP available phosphorus, AK available potassium, BD bulk density, MWHC maximum water holding capacity, PR plant richness. Plant α-diversity was represented by the Shannon index. Parameters with a significant relationship were shown in bold (*p* < 0.05).

**Table 9 microorganisms-09-01145-t009:** The relationships between the environmental variables and the soil microbial community function based on mantel test.

	Soil Bacteria		Soil Fungi		Soil Archaea	
Variables	*r*	*p*	*r*	*p*	*r*	*p*
SOC (g kg^−1^)	−0.053	0.512	−0.235	0.975	−0.174	0.872
TN (g kg^−1^)	−0.014	0.415	−0.287	0.996	−0.092	0.671
C/N	0.006	0.442	0.398	0.025	0.208	0.164
AN (mg kg^−1^)	−0.234	0.960	−0.158	0.887	−0.196	0.814
NN (mg kg^−1^)	0.423	0.0021	0.664	0.012	0.070	0.299
AP (mg kg^−1^)	0.009	0.426	−0.073	0.658	0.008	0.444
AK (mg kg^−1^)	0.547	0.012	0.307	0.060	0.151	0.221
pH	−0.013	0.495	0.344	0.029	0.106	0.269
BD (g cm^−3^)	−0.046	0.562	0.035	0.354	0.340	0.074
MWHC (*v v*^−1^, %)	−0.308	0.973	0.006	0.444	−0.045	0.533
PR	0.068	0.294	−0.021	0.488	−0.092	0.683
Tree α-diversity	0.553	0.014	0.793	0.001	0.181	0.144
Shrub α-diversity	0.369	0.034	0.699	0.002	0.049	0.293
Herb α-diversity	−0.242	0.943	−0.002	0.446	−0.103	0.705

SOC soil organic carbon, TN total nitrogen, C/N the ratio of organic carbon to total nitrogen, AN ammonium nitrogen, NN nitrate nitrogen, AP available phosphorus, AK available potassium, BD bulk density, MWHC maximum water holding capacity, PR plant richness. Plant α-diversity was represented by the Shannon index. Parameters with a significant relationship were shown in bold (*p* < 0.05).

## Data Availability

Not applicable.

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
