# Peer review of "Restoration Efficacy of Picea likiangensis var. rubescens Rehder & E. H. Wilson Plantations on the Soil Microbial Community Structure and Function in a Subalpine Area"

_microorganisms, 2021, doi:10.3390/microorganisms9061145_

Round 1
Reviewer 1 Report
The paper focuses on the changes in the structure and function of soil microbial communities in two differently aged Picea l. forests and in the Abies s. forest in a subalpine area. The topic is relevant for this journal and I find it interesting. Nevertheless, some issues need to be modified and more explained, especially (but not only) in the Materials and Methods section. The main problem is the missing replication of the study plots (if understand well based on the sampling design description) and insufficiently described vegetation conditions. The conclusions are thus strictly limited.
I am not a native speaker but I feel that the English must be checked in the manuscript.
Comments:
- Introduction is too general and a bit vague; moreover, I am completely missing the hypotheses there
- The term “primitive” coniferous forests is often used throughout the manuscript; what is meant by this term? Shouldn't there be “primeval”?
- Study sites - I miss information regarding the history of sites before planted by spruce.
- regarding the soils, there is no information about the surface organic layer (thickness or mass); was it removed before sampling?
- no information can be found regarding the vegetation (shrub and herb coverages), canopy openness, etc.
- what is the parent material of the soils? Is it the same at all plots?
- sampling design is unclear. Three stands were selected for study (2 Picea stand, one Abies). In each stand, one plot 20x20 m was established; within the plot one 5x5 m and three 2x2 m plots were distributed for vegetation description. Samples were taken as composite samples, thus for each stand was taken only one composite sample? Later it is mentioned, that there were three replications. Were samples taken on the microplots where herb layer was evaluated or not? It is very important information; if the sampling points did not located in microplots (or very near to them) it is hard to expect relations between microbiota and herb layer.
- Table 1 – check the unit for stand density,
- L157 – use “Bray-Curtis” instead of “bray-curtis”
- Figures 2 and 3 are difficult to read
- Table 7 and 9 – I do not understand; if the both spruce stands were pure in overstory than I wonder why the tree diversity indexes differ? If in the 45y old spruce stand also other trees were present then information on tree species occurring there and their proportion will be useful.
- it is difficult to judge the Discussion section because there are uncertainties in sampling design and poor description of soils as well as vegetation conditions.
Reviewer 2 Report
Keywords should be arranged alphabetically
L53-55: rather than being suggested, it was actually stated and these are facts supported by publications.
L95-97: so no replicate samples were taken? How much soil was obtained (g)?
L103: add space "density (BD) determination"
Table 1: no explanation of what "DBH" is
L107-117: analyses were done in replicates?
L135-152: any references? Readings obtained were deposited in the database?
L154-160: references?
Figure 1: in the caption explain what "others" and "PCF" are; specify at which taxonomic level the results
Table 2: instead of footnote no. 3, this information should be in the title of the table; SE should probably be given with a +/- symbol; L200: which test was used to determine the differences? standardise the number of digits after the comma in the SE notation
Figure 2: graphics are not very clear; L205-206: description is incomplete - which figure refers to what? Should be clearly signed a, b, c. What taxonomic level is involved?
Table 3: what taxonomic level?
Table 4: what test was used to determine differences?
Table 5: standardise the number of digits after commas in the SE notation. which test was used to determine the differences?
Table 7: notation of units (g kg-1); which test was used to determine the differences?
Table 8 and 9: add units to variables
Figure 4: without statistics? What units on the Y-axis?
The discussion seems too short to me and has rather few references. I.e. there are long passages without references (e.g. L366-379). Also, it relies very much on presenting results rather than comparing them with other data.
Table S1, S2 and S3: what test was used to determine differences? Add headings in the first row of each table i.e. 'order'.
Round 2
Reviewer 1 Report
Thanks to authors for the revised version of the manuscript and the responses to the comments which were incorporated. I do not have any other serious comments.
Although English was corrected I feel that there are some parts yet, which need to be improved, reformulated etc. (but I am not a native speaker), e.g.
L394 "which indicating..."
L432 - use "strongly" instead of "strong"
L439-442 - "Because we found that....., moreover,.."
Other comments:
Figure 1 and Table 2 should be placed behind the text not directly behind the title
L384 What does "Soil fertility availability" mean?
L393 but comma behind "NN"
Reviewer 2 Report
L117: space between 500 g
L124: a dot at the end of the sentence
Figure 1: further no information on what is 'others'
